# The Effect of Lysophospholipids and Sex on Growth Performance and Small Intestine Morphology in Weanling Pigs, 7–30 kg

**DOI:** 10.3390/ani14081213

**Published:** 2024-04-18

**Authors:** Sabine Stoltenberg Grove, Jacob Dall, Johannes Gulmann Madsen

**Affiliations:** 1SEGES Danish Pig Research Centre, Axeltorv 3, 1609 Copenhagen, Denmark; 2Vilofoss, Ballesvej 2, 7000 Fredericia, Denmark; 3Department of Veterinary and Animal Sciences, University of Copenhagen, Grønnegårdsvej 2, 1870 Frederiksberg, Denmark

**Keywords:** weanling pigs, lysophospholipids, lysolecithin, growth performance, gut morphology, cell proliferation

## Abstract

**Simple Summary:**

Lysophospholipids (LPL) are monoacyl derivatives of phospholipids in which one of the fatty acids has been removed by enzymatic hydrolysis and can influence lipid digestion. In this study, the effect of feeding lysophospholipids in the diet to pigs from 7–30 kg was investigated with respect to growth performance and the intestinal morphology. A total of 800 pigs were divided into two groups fed either basal diet (CON) or basal diet + 0.05% LPL (LPL), respectively, and the growth performance was measured for 42 days. At the end of the experiment, 32 pigs were euthanized to evaluate the tissue of the small intestine in the pigs. Overall, there was no statistical difference between day 1 and day 42 relating to overall gain or growth performances. However, due to the variations between the two groups starting phases, the results indicate that lysophospholipids had a positive effect on recovering from a poorer starting point. This was further reinforced by intestinal morphology analysis, which also showed no statistical difference, despite the early lag in growth performance of the LPL group.

**Abstract:**

Inclusion of lysophospholipids (LPL) has been proposed to increase growth performance in broilers and pigs, acting as emulsifiers through mixed micelle formation. The aim of this study was to investigate the effect of feeding LPL in weanling pig diets on growth performance and intestinal morphology. Eight hundred pigs (weight 6.96 kg ± SD 1.58 kg) were assigned to one of two dietary treatments, i.e., a basal diet (CON) or a basal diet + 0.05% lysophospholipids (LPL). The experimental period lasted for 42 days, and on days 40 and 41, 32 pigs in total were euthanized for intestinal tissue samples. From days 14 to 21, feed intake and average daily gain increased, as well as FCR, from days 28 to 42, in the LPL group compared with the CON group. In the overall period, no differences in growth performance were present between the groups. However, females displayed increased ADG from days 21 to 28 compared with castrates. The villous height tended (*p* = 0.051) to be lower in LPL in the proximal jejunum compared with CON. In the proximal jejunum, villus was higher (*p* > 0.01) in females, and in the distal jejunum, higher crypt cell proliferation (*p* < 0.01) and a tendency to deeper crypts (*p* = 0.064) were observed in female pigs as well. In conclusion, lysophospholipids did not increase growth performance in this study; however, the rate of recovery from a poorer starting point was noted, as growth rates recovered and increased faster in the LPL group. In conclusion, unlike the first phase, the LPL group recovered the growth from days 14 to 21 through higher feed intake and weight gain than the CON group. Eventually, the LPL group displayed improved FCR compared with the CON group from days 28 to 42. Further studies are needed to investigate whether this effect continues into the grower–finisher phase.

## 1. Introduction

Digestibility of nutrients is directly linked to growth performance, and especially in young pigs, it can be compromised due to the immaturity of the gastrointestinal tract (GIT) around weaning [1,2]. Once weaned, the digestive enzyme activity of the pig’s small intestine must adapt rapidly to be able to utilize nutrients from solid feed compared with highly digestible sow’s milk. This abrupt change in nutrient sources can lead to detrimental effects on the growth performance and health status of the pigs, as reviewed by Lallès et al. (2007) and Pluske et al. (2018) [3,4].

Dietary fat is the most energy-dense nutrient. Being non-soluble in water, it assembles into fat globules in the gastrointestinal tract. Here, pancreatic lipase and hepatic bile initiate the breakdown of larger fat molecules into free fatty acids and monoglycerides [5], which can be absorbed through the cell membrane of the epithelium. Bile is a natural emulsifier and improves the possibility of mixing hydrophilic and hydrophobic solvents and molecules through the formation of amphiphilic mixed micelles. Hydrophobic molecules can be carried inside the mixed micelles to the apical cell membrane, where they are diffused or actively transported via fatty acid-binding protein to the enterocytes, from where they can enter the circulation [6].

In young animals, the availability of bile can compromise fat digestibility [7,8]. Therefore, adding exogenous emulsifiers to the feed could potentially enhance the process of emulsification and, subsequently, the absorption of lipids. Lysophospholipids (LPL) are a group of molecules enzymatically hydrolyzed from phospholipids by the cleavage of one hydrocarbon tail. They can be found in different chemical structures, and due to their amphiphilic properties (hydrophobic hydrocarbon tail and hydrophilic polar head), they are likely to form mixed micelles [9] and be an integral part of the epithelial cell membrane [10,11]. Reynier et al., 1985 [12] and Schwarzer and Adams, 1996 [13] have demonstrated how mixed micelles containing LPL are smaller in size and therefore have a greater surface area compared with bigger micelles. Subsequently, they appear closer to the apical surface of the epithelium, which enhances the absorption of dietary fat.

The inclusion of LPL in diets for monogastric animals, mainly broilers, has been investigated in several studies in relation to improving growth performance and absorptive capacity. In contrast, most studies on weaners are scarcer and of older date [14,15,16,17,18]. A recent study conducted under non-commercial conditions found that supplementation of lysolecithin, another emulsifier, improved ADG during a period of 28 days post-weaning as well as increased ADFI from days 15 to 28 post-weaning [19]. In addition, Liu and colleagues [19] observed minor changes in ileal morphology, with a tendency toward a shorter crypt depth in the lysolecithin-supplemented group. In contrast, another recent study conducted under non-commercial conditions by Kinh et al. (2023) [20] reported that a combined supplementation of lysolecithin, synthetic emulsifiers, and monoglycerides did not elicit any improvement in growth performance during the first 56 days after weaning but did increase ME and ME intake in growing pigs (30 kg BW). However, growth performance and absorptive capacity, measured as changes in the small intestine morphology and cell proliferation, have not been investigated in piglets in the late weaning phases reared under commercial conditions.

Xing et al. (2004) [16] observed an increase in average daily gain (ADG) from days 15 to 35 and overall period (days 0 to 35), as well as an increase in final BW of weaned piglets supplemented with 0.02 and 0.10% lysolecithin, which was not followed by a change in average daily feed intake (ADFI) or feed conversion ratio (FCR). A recent study in weaners investigated the effect of LPL when energy was reduced. The inclusion of 0.1% LPL enabled the pigs to obtain similar growth performance compared with pigs fed a diet with a normal level of energy [18]. In addition, there are several examples of improvements in FCR in broilers with dietary supplementation of LPL [21,22,23].

As is the case with broilers, weaners are similarly dependent on maximizing their absorptive capacity to accommodate the high energy and nutrient requirements supporting the relative rapid growth rates characterized by young animals. For this reason, it was of great interest to evaluate the effect on growth performance and small intestinal morphology of a known absorptive accelerator in weaners reared under commercial conditions, a study that has, to the best of the author’s knowledge, not previously been performed. Thus, the overall objective of this study was to investigate the effect of LPL on the growth performance and intestinal morphology of weaners housed under conventional production conditions to evaluate its value as a feed additive in young pigs. It was hypothesized that weaners fed LPL would increase the growth performance and improve the intestinal morphology towards greater absorptive capacity throughout the experimental period.

## 2. Materials and Methods

### 2.1. Experimental Design, Animals, and Treatments

The 42-day experiment during Summer 2021 was conducted on-farm in a commercial Danish pig herd and included a total of 800 weanling pigs ((Landrace × Yorkshire) × Duroc) with an initial body weight of 6.96 kg ± SD 1.58 kg. Pigs were either weaned on the day of the experimental start (age 23 days old ± 2 days) or had been housed in a nursery pen after weaning (age 27 days old ± 2 days). The previous housing (only a farrowing pen or subsequent nursery pen) of pigs was balanced between the two groups. The pigs were sorted according to initial weight, sex, and origin, either being housed previously in a farrowing or nursery pen, and assigned to pens with 25 pigs in each pen. Thus, pigs were housed female-only and castrate-only in both the CON and LPL groups. The setup resulted in variations amongst pigs but represents the usual weaning situation and therefore the reality when working with commercial Danish pig herds. The pigs were housed in two weaner sections, with 16 pens in each section. Animals were assigned to either a control group (CON, n = 400) or a dietary treatment group with added 0.05% LPL complex to the diet (LPL, n = 400). The lysophosholipid complex was prepared from soy lecithin containing lysophospholipids with a carrier. Each group was evenly distributed between the two sections, with a total of 16 pens in each group. The health status of the pigs was monitored daily, and the prevalence of diarrhea in need of treatment from day 5 to day 12 was, respectively, 265 and 200 doses in the LPL and CON groups. During the experiment, dead pigs were removed from hospital pens. The total number of pigs removed from the dataset were 34 (CON; 16 and LPL; 18), due to either being dead or removed to hospital pens, respectively, 2.00% (0.75% and 1.25%) and 2.25% (2.00% and 0.25%) for the CON and LPL groups. Feed and water were available ad libitum during the experiment. Feed was distributed in feeders shared by two pens; hence, a double pen was used as the observational unit when the average daily feed intake and feed conversion ratio were calculated.

The flooring was in compliance with legislation, and the climate was regulated to meet the physiological needs of the animals during the experimental period. The pigs were weighed weekly at the end of each experimental week, apart from the first weighing on day 0. The pigs were weighed per pen, and the feed intake was continuously measured at double-pen level with weights installed on the feeder.

### 2.2. Diets and Feeding

The pigs were fed either a control diet or a control diet + 0.05% LPL. The addition of LPL was the only difference in the diets, and they were isoenergetic in each phase. The pigs were fed 4-phase diets using three feed recipes. The basal diets are shown in Table 1. All recipes met the nutrient requirements suggested by the Danish Pig Research Centre [24]. Pigs were fed four diets during the experimental period, i.e., diet 1 from days 0–14, diet 2 from days 15–21, diet 3 (50% of diet 2 and 50% of diet 4) from days 22–28, and diet 4 from days 28–42.

### 2.3. Experimental Procedures and Sampling

On days 40 and 41, tissue and blood samples were collected from a subpopulation of 32 pigs. Due to the experimental setup, the pigs were selected based on their visual size and appearance, aiming to select 1 pig per replicate pen with average growth performance from each of the two dietary groups. There was an uneven distribution of sexes, as 17 castrates and 15 females were used for sampling. The pigs were euthanized using captive bolts and exsanguinated in compliance with legislation. Pigs were not fasted prior to being euthanized.

The GIT was separated from the body, and the small intestine (SI) was ligated for further examination. The length and weight of the SI were measured for each pig before tissue sampling was initiated. All tissue samples were stored in a 10% formalin buffer. The SI was sampled at three sites: the proximal, medial, and distal parts of the jejunum.

### 2.4. Laboratory Analysis

All laboratory analysis was performed at the Core Facility for Integrated Microscopy (Faculty of Health and Medical Sciences, University of Copenhagen, København, Denmark). Tissue samples from the 32 pigs in formalin buffer were processed in a vacuum tissue processor (Shandon Exelsior, Kalamazoo, MI, USA) overnight and embedded in paraffin. The samples were cut into sections of 4 µm and dewaxed with Tissue Clear (Tissue-Tek^®^ Tissue Clear, Sakura Finetek Japan Co., Tokyo, Japan) and tap water. A total of 96 tissue samples were available for further examination. All analyses were performed with ZEN desk software (version 2.6) after being formatted into CZI files with the ZEISS AxioScan Z1, using ZEN Blue ver. 2.6 and a 10×/0.3 NA objective (ZEISS, Oberkochen, Germany).

#### 2.4.1. Intestinal Histology

For the morphological examination, staining with hematoxylin and eosin was completed for visualization of villi and crypts. Intact villi with belonging crypts were examined for each slide with ZEN desk software (ZEISS, Oberkochen, Germany). Villus height (VH), villus width (VW), and crypt depth (CD) were measured by the measurement tools in the ZEN desk software, and the ratio of VH to CD was calculated using the former examinations. All 96 slides were investigated, and all intact villi were measured. The average number of villi per pig (three slides) was 13, and the total number of villi in each treatment group was 212 and 218, respectively.

#### 2.4.2. Immunohistochemistry

Immunohistochemistry was performed to detect Ki67-positive nuclei as a biological marker for cell proliferation. For antigen retrieval, the sections were boiled for 15 min in EGTA buffer at pH of 9, followed by pre-incubation in 2% bovine serum albumin (BSA) for 10 min, followed by one hour of incubation at room temperature with the primary antibody from Abcam catalogue no. ab16667 anti-Ki67 antibody, which was diluted 1:1500 (Abcam, Cambridge, United Kingdom) in 2% BSA.

To amplify the reaction, the sections were incubated for 40 min with biotinylated secondary antibody immunoglobulin and goat anti-rabbit BA-1000 (Vector Laboratories, Newark, CA, USA) diluted 1:1200, adding a second layer. Afterwards, 3% H_2_O_2_ blocked the endogenous peroxidase. The third layer was formed by a preformed avidin and biotinylated horseradish peroxidase macromolecular complex (Elite ABC, code no. PK-6100, Vector Laboratories, Newark, CA, USA) for 30 min. Finally, the reaction was developed using 3,3-diaminobenzidine (SK-4105, Vector Laboratories, Newark, CA, USA) for 15 min, and counterstaining was performed with Mayers hematoxylin (Sigma-Aldrich, Saint Louis, MI, USA).

Using ZEN desk software (ZEISS, Oberkochen, Germany), training was initiated to detect Ki67 positive-stained nuclei, and after being tested and approved, the training model was run on all slides. From the analysis, the area in which positive nuclei were present as well as the total area of tissue were measured. The percentage area of constituted Ki67-immunopositive cells was then calculated and used for internal comparison between treatment groups.

### 2.5. Statistical Analysis

All data were tested for normality of residuals using the qq-plot function in R Studio version 1.4.1717 (R Development Core Team, 2020; The R Foundation, 2021, Indianapolis, IN, USA) and found to be normally distributed. All data were analyzed using R, and the results are presented as LS means with SEM. Feed conversion ratio, ADFI, and ADG were measured per double pen, as feeders were shared between two pens. The individual pig BW was based on the total pen weight divided by the number of pigs per pen. The body weight and ADG were analyzed with repeated measurements by including the corGaus function in a linear mixed model with dietary treatment and sex fixed effects, BW day 0 as a covariable, as well as a pen nested in the section as a random factor. The dietary treatment, sex, and their interaction were considered fixed effects when analyzing ADFI and FCR using a linear mixed model, with a pen nested in the section as a random effect. Intestinal weight, length, morphology characteristics, and crypt cell proliferation were analyzed using the individual pig as an experimental unit in a linear mixed model including dietary treatment, sex and their interaction as fixed effects, and pen nested in section as a random effect. The interaction was removed from the respective models because of its missing significance. A *p*-value < 0.05 was considered statistically significant, and *p*-values between ≤0.05 to ≤0.10 were considered to be a tendency. Whenever an interaction between sex and dietary treatment was present with a *p*-value < 0.05, a Tukey-Kramer post-hoc test was applied.

## 3. Results

### 3.1. Growth Performance and Treatments for Diarrhoea

No difference in body weight during the experimental period was observed (Table 2). On days 0–14, pigs in the CON group had significantly higher ADG and lower FCR compared with LPL pigs (Table 3). In the period from 14 to 21 days, the LPL pigs had both higher ADG and ADFI compared with CON pigs, but the FCR was still better in CON pigs (Table 3). From days 28 to 42, FCR was significantly lower in the LPL group (Table 3). Overall, there was no effect on either ADG, ADFI, or FCR when assessing the whole period from day 0 to day 42 (Table 3). There were only a few differences between females and castrates. From days 21 to 28, females had a greater ADG compared with castrates, and from days 28 to 42, females had an increased ADFI. The sex did not influence the overall results for the entire period (Table 3). During the experimental period, 200 treatments against diarrhea were registered for the CON group and 264 treatments for the LPL group. In total, the number of days of treatment per pig during the experimental period was 0.50 days for the CON group and 0.66 days for the LPL group.

### 3.2. Intestinal Histology and Immunohistochemistry

In the proximal part of the jejunum, there was a tendency to have a greater VH of CON compared with LPL pigs (*p* = 0.051) (Table 4). The VH also differed between females and castrates, with females displaying the longest (*p* < 0.01) villi. Also, the CD was longer (*p* = 0.04) in females compared with castrates. In the medial part of the jejunum, no differences were present between either dietary groups or sexes. In the distal part of the jejunum, there was a tendency (*p* = 0.064) towards longer CD in females compared with castrates, and cell proliferation, as assessed by the percentage area of Ki67-immunopositive cells, was greater (*p* < 0.01) in females compared with castrates. The dietary treatment had no effect on cell proliferation in any of the SI segments.

### 3.3. Weight and Length of the Small Intestine Intestinal

There were no differences in the small intestine characteristics (Table 5) between LPL and CON pigs in addition to sex.

## 4. Discussion

### 4.1. Growth Performance

The effects of adding LPL to diets for growing animals have been investigated in a great number of studies. Several studies have focused on investigating the effect on growth performance in broilers, in which it has been reported that LPL enhances growth performance in mainly young chicks [25,26,27]. Zhang et al. (2011) [25] reported improved ADG in broilers on day 21 after hatching when supplemented with 0.05% lysophosphatidylcholine (LPC). An increased ADG was also observed by Zhao and Kim (2017) [27] on day 14 after hatching when feeding either 0.05% or 0.1% LPL in the diet. The beneficial improvement followed by LPL supplementation is likely associated with insufficient secretion of both hepatic bile and pancreatic lipase in the early period of a chick’s life [7,28]. The digestibility of fat increases rapidly for unsaturated fatty acids and more slowly for saturated fatty acids; hence, an enhancement in growth performance is expected mainly in the early period for both unsaturated and saturated fatty acids [7]. In addition, there are studies reporting positive effects on growth performance in the later growth phases as well. Gheisar et al. (2015) [21] found greater ADFI and ADG in the last phase from days 21 to 35 in broilers fed 0.08% lysolecithin compared with a basal diet. Feed conversion ratio improved from day 7 and onwards, but there were no differences between the groups during the first week. Similarly, Boontiam et al. (2017) [22] reported increased FCR, ADFI, and ADG from days 25 to 35 when including LPL supplementation in the diet. Chen et al. (2019) [29] observed increased ADG from day 21 to day 42 with the inclusion of 0.75% LPL, arguing the improvements were likely caused by an increase in apparent metabolizable energy. These effects observed in the later phases could be due to the incorporation of LPL into the SI cell membrane, which changes the physiological properties of the cell membrane [22]. Changes in fluidity and subsequently nutrient permeability via alteration of protein channels might influence the uptake and nutrient digestibility across the enterocyte cell membrane [30,31,32]. However, in contrast to broilers, which are prone to responding to relatively small changes in feed composition during the first weeks of life, pigs display a greater ability to adapt to minor variations in nutrient content.

To the best of the authors’ knowledge, no previous studies have focused on the effect of LPL supplementation on weanling pigs reared under commercial conditions from weaning to 30 kg BW. During the overall experimental period, no improvements in growth performance were observed. These observations are in accordance with previous studies by Papadopoulos et al. (2014) [17] and de Rodas et al. (1995) [15], who also did not observe any effects on growth performance in weanling pigs. However, differences in growth performance were found within the phases between days 0 to 14 and days 14 to 21, as well as days 28 to 42. From days 0 to 14, LPL compared with CON pigs did display lower ADG and poorer FCR, as well as a tendency to a greater ADFI. An increase in ADFI has previously been explained by differences in palatability, leading to a greater ADG without affecting FCR [33]. However, the increased ADFI in the current study was only observed from days 14 to 21, suggesting that this effect was likely caused by physiological changes in the GIT rather than palatability, which is assumed to be similar in all diets provided during the experimental period. Another explanation could be a compensatory growth mechanism, as the LPL-supplemented pigs experienced a growth check during the first weeks of the experiment, which could mask the possible effect of the LPL supplementation [34]. Finally, and most interestingly, the LPL group displayed improved FCR compared with the CON group from days 28 to 42. This suggests that the LPL did in this period improve digestibility, potentially through increased utilization of lipids. It could be speculated that the decrease in ADG the first 14 days after weaning might have been related to slightly poorer health, which was to some degree supported by the numerical difference in diarrhea prevalence between the two groups favoring the CON pigs. Hence, the potential effect of LPL supplementation could be diminished or completely absent during this period. It is commonly recognized that weaning stresses animals and can have a negative impact on the morphology and function of the small intestine [35]. However, curiously, the LPL group did in fact tend to eat more than the CON group the first 14 days after weaning, suggesting that the LPL group accommodated well during the transition to weaning.

A third explanation for the missing effect in the earlier phases could also be related to the time of supplementation. As previously discussed, LPL supplements have shown great potential during the early phase of broilers, an effect that might be attributed to the chick’s inability to synthesize lipase, thus hampering fat digestibility. A physiological constraint has also been found in piglets, which, in contrast, are compensated by lipase present in colostrum ingested in the immediate hours after birth. In this regard, a few studies have observed improvements in suckling piglet performance when sows were fed LPL during late-gestation and lactation [36,37], which presumably relates to enhanced utilization of nutrients in sows leading to greater milk yield and improved nutrient composition, thus improving litter growth performance.

### 4.2. Morphology of the Small Intestine

The intestinal brush border is covered with villi structures, which increase the surface area of the epithelium by villi and microvilli, thereby enlarging the area for nutrient absorption [38,39]. A previous study investigating 0.05% LPL supplementation in broilers fed a low-energy diet observed an increase in villi height and the VH/CD ratio of the jejunal but not the duodenal segment, suggesting a segment-specific and not a general effect on the small intestine [22]. To the best of the author’s knowledge, no studies have evaluated the effect of LPL supplementation on intestinal morphology in weanling pigs, but it has been reported to increase the VH and VH/CD ratio in broilers [22,40,41]. In the present study, only minor alterations in the gut morphology were elicited by LPL supplementation, as VH tended to decrease in the proximal part of the jejunum. This decrease is likely not a consequence of the detrimental growth period observed the first two weeks after weaning, as samples of the SI were collected on days 40–41 post-weaning. In this later phase of weaning, pigs have probably established a regular level of feed intake, and the observed changes in gut morphology likely relate more to a temporary change in a smaller part of the SI area.

The jejunal segment serves as the main area of nutrient absorption in the small intestine, and thus its morphology and cell proliferation capacity were evaluated. The percentage area of Ki67-immunopositive cells in the mucosa is where Ki67 is regarded as a reliable marker for mitotic cell division [42]. A higher percentage area of Ki67-immunopositive cells has been observed previously in suckling piglets when sows were fed an LPL-supplemented diet and in broiler-fed diets with LPL inclusion [23,36]. However, no differences in the percentage area of Ki67-immunopositive cells were observed in this study. Despite these results, growth performance did not confirm this detrimental effect of negative alterations in VH, as growth performance was increased at the time of tissue sampling in the LPL-supplemented group, hence nutrient permeability might be increased despite shorter villi in the proximal jejunum. The percentage area of Ki67-immunopositive cells represents cell proliferation, but no difference between treatments regarding this parameter was found in the jejunal epithelium. Hence, it can be speculated that smaller villi do not necessarily conflict with growth performance if the shedding and renewal of absorptive cells function adequately.

## 5. Conclusions

In conclusion, weanling pigs fed an LPL-supplemented diet did not improve growth performance during the overall experimental period. From days 0 to 14, LPL-supplemented pigs displayed a lower ADG compared with non-supplemented CON pigs, which contrasted with the later phases from days 14 to 21, where LPL-supplemented pigs displayed improved ADG as compared with non-supplemented CON pigs. However, in both phases, FCR was still better in the CON group compared with the LPL group, and it cannot be excluded that the improved ADG was mainly an effect of compensatory growth, possibly assisted by the LPL supplementation. Furthermore, the LPL group displayed improved FCR compared with the CON group from days 28 to 42. On the histological level, the VH was decreased in the proximal part of the jejunum in the LPL compared with the CON group, where the contribution of the proximal part to the total absorption capacity relative to the medial part is small and therefore likely not affecting ADG largely. Bearing in mind that the experiment was conducted under practical conditions, LPL supplementation did show some potential to improve the growth of weanling pigs in the period from days 14 to 21 after weaning.

## Figures and Tables

**Table 1 animals-14-01213-t001:** Ingredients and calculated nutrient composition of the basal (CON) and LPL-supplemented (LPL) diet.

	Diet 1	Diet 2	Diet 3	Diet 4
Ingredients, %				
Barley	25	25	30.5	36
Wheat	38.8	48	41.6	35.1
Fishmeal	4	2.5	1.3	0
Soy Oil	2.2	3	2.1	1.2
Soybean meal	-	9	15.9	22.7
Premix	30 ^1^	12.4 ^2^	8.7 ^3^	5 ^4^
Calculated energy and nutrient composition				
Metabolizable energy, MJ/kg	13.6	13.2	13.05	12.9
Crude protein, g/kg	142.0	148.2	152.0	155.9
Amino acids, SID g/kg				
Lys	11.2	11.6	11.7	11.8
Met	3.9	3.8	3.8	3.7
Thr	7.3	7.3	7.3	7.3
Trp	2.5	2.5	2.5	2.5
Val	7.7	7.7	7.7	7.7
Ile	5.9	5.8	5.9	6.0
Leu	10.9	10.8	10.9	11.0
His	3.2	3.4	3.6	3.7
Phe + Tyr	11.8	11.9	12.1	12.3
Minerals				
Ca, g/kg	5.9	8.1	8.2	8.3
P, g/kg	6.3	5.8	5.4	5.1
Mg, g/kg	1.5	1.4	1.4	1.5
Na, g/kg	3.0	2.4	2.5	2.5
Cl, g/kg	5.3	4.4	4.5	4.6
K, g/kg	5.8	5.9	6.8	7.6
Mn, mg/kg	50	45	44	43
Fe, mg/kg	187	170	165	161
Cu, mg/kg	135	135	107	80
Zn, mg/kg	100	100	105	110
Se, mg/kg	0.35	0.35	0.35	0.35
I, mg/kg	0.25	0.23	0.22	0.22

^1^ Premix containing/kg: Benzoid acid 0.5%, Calcium Formate 0.3%, Prowean 200 mg, Luctarom (Vanilla/Cheese) 500 mg, Luctarom (Red Berries) 200 mg, Ronozyme Multigrain 100 mg, and Lactose, 60 g. ^2^ Premix containing/kg: Benzoid acid 0.5%, Calcium Formate 0.3%, Calcium Lactate 0.1%, Prowean 156 mg, AntiTox S 0.5%, Luctarom (Vanilla/Cheese) 250 mg, Luctarom (Red Berries) 125 mg, and Ronozyme Multigrain 100 mg. ^3^ Premix containing/kg: Benzoid acid 0.5%, Calcium Formate 0.2%, Calcium Lactate 0.1%, Prowean 78 mg, AntiTox S 025%, Luctarom (Vanilla/Cheese) 125 mg, Luctarom (Red Berries) 63 mg, and Ronozyme Multigrain 150 mg. ^4^ Premix containing/kg: Benzoid acid 0.5%, and Ronozyme Multigrain 200 mg.

**Table 2 animals-14-01213-t002:** Bodyweight of pigs fed a basal (CON) or lysophosholipid-supplemented (LPL) diet from days 0 to 42 after weaning.

Body Weight ^1^	CON	LPL	Female	Castrate	SEM	*p*-Values	
						Group	Sex
No. of Pens	16	16	17	15			
Day ^2^ 0	7.01	6.90	7.09	6.82	0.05	0.78	0.47
Day 14	9.73	9.36	9.72	9.36	0.16	0.68	0.45
Day 21	12.34	12.25	12.48	12.11	0.16	0.86	0.60
Day 28	16.31	16.14	16.41	16.04	0.16	0.97	0.31
Day 42	26.68	27.06	27.05	26.68	0.16	0.66	0.26

^1^ The individual pig BW was based on the total pen weight divided by the number of pigs per pen. ^2^ Experimental days after weaning. Pigs were either 23 days old (±2 days) or 27 days old (±2 days) at the start of the experimental period, depending on previous housing (farrowing or nursery pens).

**Table 3 animals-14-01213-t003:** Average daily gain (ADG), average daily feed intake (ADFI), and feed conversion ratio (FCR) of pigs fed a basal (CON) or lysophosholipid-supplemented (LPL) diet from days 0 to 42 after weaning.

Performance Parameters		CON	LPL	Female	Castrate	SEM	*p*-Values	
							Group	Sex
No. of Pens		16	16	17	15			
Day ^4^ 0–14	ADG ^1^	195	174	187	183	8.94	0.01	0.59
	ADFI ^2^	240	274	265	249	8.13	0.06	0.35
	FCR ^3^	1.27	1.57	1.40	1.43	0.03	<0.01	0.52
Day 14–21	ADG	373	411	388	396	8.20	0.05	0.64
	ADFI	396	491	441	446	11.31	<0.01	0.88
	FCR	1.07	1.20	1.41	1.12	0.01	0.02	0.75
Day 21–28	ADG	567	556	589	534	11.03	0.59	0.01
	ADFI	735	761	780	716	18.92	0.56	0.16
	FCR	1.29	1.37	1.33	1.33	0.01	0.13	0.99
Day 28–42	ADG	671	704	701	673	3.92	0.17	0.24
	ADFI	1110	1100	1140	1070	13.26	0.50	0.06
	FCR	1.66	1.56	1.62	1.59	0.01	<0.01	0.72
Day 0–42	ADG	444	452	458	439	7.21	0.57	0.17
	ADFI	630	663	670	631	11.84	0.50	0.30
	FCR	1.41	1.45	1.44	1.42	0.01	0.70	0.75

^1^ gram/day. ^2^ gram/day. ^3^ kg feed/kg gain. ^4^ Experimental days after weaning. Pigs were either 23 days old (±2 days) or 27 days old (±2 days) at the start of the experimental period, depending on previous housing (farrowing or nursery pens).

**Table 4 animals-14-01213-t004:** Villous height, villus width, crypt depth, and cell proliferation measurements of pigs slaughtered on day 42 post-weaning and fed a basal (CON) or lysophosholipid-supplemented (LPL) diet from days 0 to 42 after weaning.

Observation	CON	LPL	Female	Castrate	SEM	*p*-Values	
						Group	Sex
No. of Animals	16	16	17	15			
Proximal jejunum, µm							
Villus height	314	281	330	265	2.33	0.05	<0.01
Villus width	137	153	148	143	1.79	0.17	0.63
Crypt depth	275	261	286	249	5.76	0.33	0.04
VH:CD ^1^	1.15	1.13	1.19	1.09	0.02	0.74	0.25
Crypt cell proliferation ^2^	7.95	7.85	8.03	6.68	0.16	0.66	0.26
Medial jejunum, µm							
Villus height	345	319	325	338	3.24	0.28	0.59
Villus width	155	162	163	153	1.46	0.57	0.39
Crypt depth	281	276	278	280	3.25	0.72	0.87
VH:CD ^1^	1.25	1.17	1.18	1.23	0.01	0.43	0.59
Crypt cell proliferation ^2^	7.11	7.29	7.35	7.05	0.12	0.86	0.77
Distal jejunum, µm							
Villus height	265	290	277	278	4.74	0.25	0.97
Villus width	137	135	132	140	1.41	0.91	0.50
Crypt depth	280	289	297	272	2.78	0.55	0.06
VH:CD ^1^	0.97	1.01	0.94	1.04	0.01	0.56	0.25
Crypt cell proliferation ^2^	7.26	7.22	8.01	6.47	0.10	0.97	<0.01

^1^ Villus height/crypt depth ratio. ^2^ Percentage area of Ki67-immunopositive cells.

**Table 5 animals-14-01213-t005:** Weight and length of the small intestine of pigs slaughtered on day 42 post-weaning, fed a basal (CON) or lysophosholipid-supplemented (LPL) diet from days 0 to 42 after weaning.

	CON	LPL	Female	Castrate	SEM	*p*-Values	
						Group	Sex
No. of Animals	16	16	17	15			
Weight of intestine, g	1126	1176	1178	1124	36.5	0.38	0.28
Length of intestine, m	15.80	15.80	15.60	16.00	0.30	0.90	0.33

## Data Availability

The original contributions presented in the study are included in the article material, further inquiries can be directed to the corresponding author.

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
