# Peer review of "The Effect of Lysophospholipids and Sex on Growth Performance and Small Intestine Morphology in Weanling Pigs, 7–30 kg"

_animals, 2024, doi:10.3390/ani14081213_

Round 1

Reviewer 1 Report (Previous Reviewer 1)

Comments and Suggestions for Authors

Line 15 Lysophospholipids instead LPL

Line 22 Although you wrote the recipe for the LPL group in the Abstract, is it not easy for the reader to understand if the Simple Summary presents the LPL group directly?

Line 27 LPL suggests change to LPL group

Table 4 provides very detailed information on the crypts and villi of the various intestinal segments under different treatments, but I would suggest that some pictures could be provided to support this. For example pictures of immunohistochemistry of the intestine.

In Table 5 you test blood glucose levels, but I don't see a reason for testing blood glucose in your  Introduction and there is no description of the blood glucose results in the discussion. Based on the fact that you are using LPL as a supplement, then why not test for TG and T-CHO in the piglets' blood?

Author Response

Dear reviewers, thank you very much for your thorough review. Please find answers to questions in rebuttal and revised manuscript attached.

Reviewer 2 Report (Previous Reviewer 2)

Comments and Suggestions for Authors

This manuscript mainly discussed the effect of lysophospholipid supplementation and the effect of sex on growth performance and intestinal morphology of weaned pigs. The main findings of the manuscript were that the inclusion of lysophospholipids at 0.05% neither improved nor impaired the cumulative growth performance of weaned pigs in a 42-day nursery study in a commercial setting. The authors also reported few morphological changes in the proximal jejunum by lysophospholipids supplementation.

Here are my detailed comments on the revised manuscript:

1.      The sex effect on growth performance and small intestine morphology in weanling pigs, 7-30 kg was present in the manuscript’s title. However, there is no information to support why determining the sex effect on growth performance or its interaction with lysophospholipids in the study is important.

2.      Immunoglobulin A was removed from the study, so keywords should be revised.

3.      Ln 76-77: The authors stated “absorptive capacity measured as changes in the small intestine morphology and cell proliferation has not been investigated in weaners.” Please consider revising for accuracy. You should be able to see other publications regarding the effects of dietary lysophospholipids on nutrient digestibility and intestinal morphology.

4.      Ln 78-80: It is better to state the response is in which species

5.      Ln 85-87: The authors stated weaners and broilers are similar. Please consider revising for clarity.

6.      Ln 87-89: How do lysophospholipids used in this study differ from lysophospholipids used in previous publications? Please explain; otherwise, the effect of lysophospholipids have been investigated previously, then the current description in the introduction is misleading.

7. Ln 125: The discrepancy between Table 1 and Section 2.2 is that seeing 4 phase feeding in Table 1, but “three feed recipes” was stated in Section 2.2. Aren't four feed recipes? Also, the nutrient requirements suggested by the Danish Pig Research Centre (Tybirk et al., 2021). Tybirk et al. (2021) research is not reported in the reference. It seems protein level and digestible Lys are very low. Are there any reasons to formulate this type of nursery ration?

8.      Ln 170: For the morphology, please define how you measured VH, VW, and CD. The average number of villi per pig (three slides) was 13, which means only about 4.3 villi per slide (i.e., per segment of the jejunum) was measured. This number also seems very low and might be not representative enough to measure morphological characteristics. Researchers will randomly select 5-10 fields and each field should have a few villi and crypt to measure under a specified magnification.

9.      Ln 200: Are only body weight and ADG analyzed with repeated measurements? Please describe how ADFI and FCR were analyzed.

10.   Ln 214: The authors reported FCR as FUgp/kg gain, which is not often used in scientific writing to report FCR. Please consider changing FCR calculation the per unit of feed intake to per unit of product.

11.   Ln 217-219: Please consider revising “FCR was still improved in CON” to FCR was still better in CON.

12.   Ln 259: Please define LPC

13.   Ln 263-266: Please consider revising for clarity. The reference [7] Krogdahl A. (1985) stated that “Addition of bile salts, lipase or phospholipids to chick diets improves the digestibility of animal fats, demonstrating that lipid digestive processes are not fully functional in the very young.” Hence, the effect on saturated fatty acids should be expected instead of unsaturated fatty acids.

14.   Ln 273-274: The sentence does not fit here. If there is an increased need for maintenance energy with LPL supplementation, improvement in growth performance is unlikely. 

15.   Ln 280-283: This statement is questionable. Weaned pigs had poor adaption to solid feed after weaning, which is well reported. Also weaning body weight and end-of-nursery body weight are a predictor for performance up to slaughter, so piglet responding additive should be greatly influencing the later phase.

16. Ln 302-305: The results were explained by a poorer health condition; then, I would think it is necessary to conduct statistical analysis to compare the treatment of diarrhea. It seems odd that LPL group had a greater diarrhea incidence than control but the dead and removals were similar between two treatments.

17.   Ln 304 and 308: The sentence is repeated.

18.   Ln 311-312: The authors presented that LPL consumed more feed, suggesting LPL group accommodated well after weaning. However, the authors previously presented a poor health status in LPL group, so the gain was poorer than control group. Typically, if early intake is not good, then gain is suffered and health in general is not good as well for weaned pigs.

19.   Ln 329-331: A thorough literature review is needed for a better discussion. There are studies that have evaluated the effect of LPL supplementation on intestinal morphology. Please check more journals, like the Journal of Animal Science.

20.   Ln 353-356: The FCR is still better in control compared with LPL treatment even though there is an improvement in ADG from day 14 to 21.

21.   Ln 362-364: cumulative performance was not affected by LPL under practical conditions, so a specific period, like d 14-21, is needed to present in the sentence “LPL supplementation did show some potential to improve growth of weanling pigs.” 

Author Response

Dear reviewers, thank you very much for your thorough review. Please find answers to questions in rebuttal and revised manuscript attached.

Reviewer 3 Report (New Reviewer)

Comments and Suggestions for Authors

Review of “ The effect of lysophospholipids and sex on growth performance and small intestine morphology in weanling pigs, 7-30kg”

Overall, an interesting experiment and topic looking at the impact of lysophopholipid inclusion in feed to piglets in the post wean period up to 42 days post. I commend you for completing a trial under production conditions, it has its own challenges but results in a higher application for industry.

There are a few questions I have for the authors regarding their methodologies and clarity in discussion points.

You state in your abstract and throughout your paper in reference to the LPS group “recovery from a poorer start point”. But you also state no difference in body weight, Table 2 shows a 100g difference between your control animals and LPL animals on day 0. Your LPL piglets were slower to recover which could be diet based and could also have some effect of a slightly lighter start weight. If you accounted for start weight in your statistical modelling this could be determined with more certainty.

Methodology:

You state in your methodology that the piglets were sourced from two different housing prior to the experimental start. Although you have tried to account for that by balancing between the two locations, you have a large amount of age variation and also potential adaptation to the diet effect. Have you separated this effect in your statistical analysis? I would assume that piglets from the nursery would have better growth as they would already be on a solid diet for ~5 days. If possible, it would also be good to know how litter was treated or if it could be as this has a very big impact on post wean performance as their early life establishment is somewhat “unique” to their sow and farrowing pen. This pen of origin would also be interesting to see if that had any trends for the diarrhoea occurrence.

Lines 114 you use the term deserted. I believe a better choice would be removed from the data set, if I have your meaning correct.

A visual representation of your pen design would be beneficial to show the food sharing and potential contact of pigs between pens. Did you find neighbouring pens had higher diarrhoea occurrence or a particular room?

Line 120: floorage would be better as “flooring”.

Table 1:  do you need the different columns for control and LPL if they are all the same numbers? I would reduce it down to one column per diet since you have already mentioned the only difference is the inclusion of LPL. Your first footnote would also work well in the text above the table. That explains the diet phases more clearly. You also have some random spaces and miss alignments in your first, 4, 7 and 8th columns.

Line 152, you mention that you selected your euthanized pigs on visual appearance. Using what criteria? Visual is biased, why not random selection? Your intestinal morphology seemed to come out relatively good, was this because you may have selected all high health animals?

For your gut section collections, did you use cassettes or another method to stop the curling of the intestinal sections? This can sometimes influence appearance of villus making interpretation difficult.

Why jejunum? Why not ilium or duodenum ?

STATISTICS: Did you look at any other factors in your statistical modelling? Origin, initial body weight, age etc? or an interaction effect considering you found sex to be different for a number of measures, did the diet have a greater effect on one sex over the other?

Results:

Intestinal histology and immunohistochemistry: You quote difference in VH and CD but not the VH:CD ratio in text which is more telling. As the ratio arguably plays a greater role in gut health. Table4:  Considering that you saw an increase in control VH and females, have you checked if this was a particular pen or factor skewing the data? The interaction effect if there is any may help to pull this apart.

In your results and methods, you do not report the number of villi counted. It would be interesting as to if one group had a greater number than another and if accounting for this makes any difference to your interpretation.

Discussion:

Lines 254 -284: although all interesting information, it is repetitive of the introduction and is not discussed in context of the results of this study. I would suggest starting your discussion with some of your own results and then showing the relevance to these studies throughout. Currently it reads more like a literature review.

Line 291-292: You quote that FCR for day 0-14 was not affected by dietary treatment. Your table shows it is higher in LPL. Increase in ADIF was also seen as a trend in day 0-14 not just 14-21.

Line 298: you note that only the LPL pigs experienced a growth check. As theoretically (and seen in other studies) a growth check could have occurred between day 0-7 for control animals especially as they had the faster suspected recovery.

I would be cautious about your diarrhoea and poor health start claims unless you have investigated if there is an origin effect and also the 64 pig difference in diarrhoea occurrence could be due to many interactive factors.

Morphology of small intestine: Although you state no studies effect of LPL on intestinal morphology in weaning, there are plenty with other food supplements and other studies which report VH and CD. How does your compare to other studies. Would this indicate that your pigs were all relatively healthy anyway ? especially your subset, did you unconsciously select healthy pigs for your subset?

Conclusion:

You quote your ADG a lot in your conclusion, I would also include the FCR being higher in the day 0-14 of the study, as this provides a good contrast to the improved FCR in the later stages of the study.

Your sex effect appears a lot throughout your results, if you are able to determine if it is a sex by treatment effect this would be valuable for the conclusion as in some production systems it is normal to house sexes separately and as such feeding them differently could be an approach and potentially more cost effective for the producer.

References:

Check your formatting. Some of your DOI are full https links and others are just the DOI.

Comments on the Quality of English Language

English language use is fairly good. The structure of discussion etc needs some improvement for flow and readability. 

Author Response

Dear reviewers, thank you very much for your thorough review. Please find answers to questions in rebuttal and revised manuscript attached.

Reviewer 4 Report (New Reviewer)

Comments and Suggestions for Authors

The manuscript is generally well-written and adheres to scientific principles. The discussion is also well-executed, with relevant and current references. However, the work has some shortcomings that need to be addressed and several errors that require correction, especially in materials and methods section.

L130 Incorrect reference format.

Table 1: To enhance readability, it is suggested to specify in the table header the periods during which the respective diets were administered, instead of including this information in the table footnote.

The description of experimental replicate pens lacks clarity. Were they formed based on dietary treatment and piglet sex, implying the existence of female-only or castrate-only control pens, and female-only or castrate-only LPL pens, as suggested by subheadings in Tables 2-3? If so, why wasn't a (treatment) x (sex) interaction measured for performance data?

L153 Please specify if the selection of pigs included one pig per replicate pen. Were pigs fasted before euthanasia? If so, specify the fasting period.

L156 Provide detailed information about the device used for blood glucose measurements.

L166 Specify the supplier of Tissue Clear (Sakura?).

L180 Clarify whether boiling was performed in a water bath, pressure cooker, or microwave.

L181 Verify if EGTA-buffer was used. Typically, an EDTA-buffer (Tris/EDTA-buffer, pH 9) is used for antigen retrieval, not EGTA-buffer.

L187 Blocking of endogenous peroxidase is typically performed at different steps in the IHC procedure (before or after antigen retrieval, and sometimes after primary antibody incubation), but blocking after secondary antibody incubation is very unusual. Please provide a reference for this IHC protocol.

Tables: As tables need to be self-explanatory, please add in the table footnote that 'day' refers to experimental days, not the age-days of piglets (and provide information about the age of piglets at the start of the experiment). Also, explain all abbreviations used in the tables.

L241-242, L337-342  he applied measurement methodology and the results demonstrate the percentage of the area showing Ki67-immunopositive cells, not the percentage of Ki67-positive nuclei/cells. These are distinct measurements.

Data in Table 5 were measured at the animal level; thus, a (treatment) x (sex) interaction could also be included in the statistical model.

Conversely, as no significant (treatment) x (sex) interaction was found, it is recommended to remove it from the statistical model and rerun the statistical analysis, highlighting this fact (i.e., rerunning the statistical analysis due to the lack of interaction significance).

L284, L328 “authors’ “

L304 Statistical analysis should be conducted to confirm differences in diarrhea prevalence between experimental groups. Without this analysis, the statement remains an unverified hypothesis.

L331 “small intestine” or “jejunum”, not the (whole) gut.

L337-349 It would be beneficial to specify the primary functions of the jejunum (like absorption of monosaccharides, fatty acids, amino acids, some water-soluble vitamins, and minerals).

References are not formatted in accordance with journal requirements. Please revise.

Comments on the Quality of English Language

Minor grammar corrections are required.

Author Response

Dear reviewers, thank you very much for your thorough review. Please find answers to questions in rebuttal and revised manuscript attached.

Round 2

Reviewer 2 Report (Previous Reviewer 2)

Comments and Suggestions for Authors

Here are some comments on the revised manuscript:

1.      Ln 78-79: The sentence is not complete

2.      Ln 79-82: Please revise the sentence for clarity.

3.      Ln 84-86: Please revise the sentence for clarity. The entire sentence did not reveal what treatments in Xing et al. (2004) study resulted in the responses you reported.  

4.      Ln 91-93: Please revise the sentence for clarity.

5.      Ln 118-120: Please report the total concentration of lysopholipids in lysopholipid complex as it is not 100% lysopholipids and contains lysophospholipids and carrier. 

6.      Ln 137-138: I would state that the pigs were fed 4-phase diets using three feed recipes.

7.      Ln 141-143: Please consider moving this to section 2.2 Diet and Feeding.

8.      Ln 152: blood sample for analysis were not reported, so please consider revise the sentence for clarity

9.      Ln 201: Typo “Ki67-immunopositive celss »

10.   Ln 307-310 : Please further explain how feed intake response is caused by physiological changes in the GIT?

11.   Ln 351-353: Please revise the sentence for clarity.

Author Response

Dear reviewer, thank you very much for your thorough review. Please find answers to questions below and revised manuscripts attached.

This manuscript is a resubmission of an earlier submission. The following is a list of the peer review reports and author responses from that submission.

Round 1

Reviewer 1 Report

Comments and Suggestions for Authors

Manuscript ID: animals-2213805

Title: The effect of lysophospholipids and sex on growth performance and health status in weanling pigs, 7-30 kg.

In this manuscript, the authors investigated the effect of feeding lysophospholipids on growth performance and the intestinal morphology,  didn't review the the level of immunoglobulin A as author described. Furthermore, the author just described the phenotype, should be further explored the machinism. The results of intestinal histology and immunohistochemistry should add the images were captured, to more intuitive presentation of results. The conclusion should highlight the significance to production practice throough this study.

1.      Line 111-112: Why the desertion due to dead or removal to hospital pens were increased 0.3% in the LPL group than the CON group? The rate of desertion due to dead and removal to hospital pens in two groups, respectively?

2.      Line 216: Should set text-indent.

3.      The layout of line number should modify at Table 2.

4.      What are the reasons for high ADFI? What season is the experimental conduct? 

Author Response

1. Line 111-112:

Why the desertion due to dead or removal to hospital pens were increased0.3% in the LPL group than the CON group? The rate of desertion due to dead and removal to hospital pens in two groups, respectively?

We are not exactly sure about the question. But if it relates to distribution between dead and hospitalized in the two groups it has now been added. The cause of death and hospitalization was apparently not noted.

2. Line 216: Should set text-indent.

Corrected.

3. The layout of line number should modify at Table 2.

We are not sure what is meant here?

4. What are the reasons for high ADFI?

Which specific values are the reviewer questioning.

What season is the experimental conduct?

Summer. Now included under point 2.1.

Reviewer 2 Report

Comments and Suggestions for Authors

This manuscript mainly discusses the supplementation of lysophospholipids on growth performance and jejunal morphology of weaned pigs. The main contribution of the manuscript is that the inclusion of lysophospholipids at 0.05% did not negatively affect any growth parameters of weaned pigs fed diets containing metabolizable energy between 3250 kcal/kg to 3083 kcal/kg during the nursery period in a commercial environment. Although the authors observed some morphological changes in the jejunum, the response is mainly induced by sex but not by the supplementation of lysophospholipids. The authors further analyzed the IgA concentrations in the jejunal mucosa, but unfortunately, they were unable to compare the result between treatments due to technical issues. The authors also observed a few differences in some parameters between barrows and gilts, but the interactions between sex and lysophospholipids were not reported in all data. Finally, the authors stated the improvement in feed efficiency in the last phase of the nursery period by adding lysophospholipids as a part of their conclusion.

Here are my major comments:

1. In the introduction, the authors acknowledged that previous results on the application of lysophospholipids are inconsistent due to the dose and source of dietary fat, but they did not design the study to evaluate different inclusion rates of lysophospholipids with multiple lipid sources or at least with animal fats. Instead, they repeated as a previous study to test the supplementation of lysophospholipids at a single dose with soybean oil, which is already known for its great digestibility in young animals.

2. Ln 78-83: the authors provided a couple of studies that evaluated the supplementation of lysophospholipids on growth performance of poultry and swine. They should indicate that the study is based on broiler or swine trials. For example, “There are several examples of improvements in FCR in weaners with dietary addition of LPL [21, 22, 23].” References 21, 22, and 23 are all from poultry, but the authors stated the improvements in FCR in swine. This is considered overgeneralization in scientific writing.

3. Ln 89-93: The significance of this research should be clearly stated. Providing information about tissue morphology and pigs raised under commercial environments is interesting, but how does this information benefit the field or why the research should be conducted in this approach?

4. The study used 800 weaned pigs with an initial body weight of 6.96 kg ± 1.58 kg. This creates a coefficient of variation of almost 23%. The high CV suggests no selection on the allotment and potentially with a considerable variation between pens within the treatment that might compromise treatment effects even though the authors balanced the initial body weight with similar average body weight between treatments in the beginning.

5. For sampling, 1 pig per pen were randomly selected. As each pen has 25 pigs at the allotment of the current feeding study, the selection of 1 pig without any criteria is hardly a sufficient number to generate meaningful results. Some researchers can argue that your findings are from sampling error.

6. The experimental diets were 4 phases in table 1, but this differs from what their description in the section 2.2. Also, very little detail is provided in diet composition. While diets are isocaloric with equal SID Lys, the available phosphorus, Ca:P ratio, NDF are missing. Also, no trace minerals content was reported. Please define FU/kg in the table. Analyzed nutrients should also report in the table.

7. For the morphology, the intact villi was measured with the number of villi per slide from 1-17, which is a very small number. Normally, researchers will randomly select 10 fields under 100 × magnification or other magnification to measure villi and crypt. No relevant information was provided in the methods.

8. What is the reason for not counting the number of Ki-67 positive cells per field and counting the total enterocytes per field to calculate the percentage of cell proliferation?  

9. Although the authors stated that supplementation of lysophopholipids can have positive impact on performance by recovering from a poorer starting point, the intake of pigs in lysophopholipid group is actually better than control with a similar growth rate in the first 4 weeks. For me, this suggests that those pigs did not have a hard time post-weaning because pigs were eating well. Hence, it is hard to convince readers of Animal Science field to accept the author’s statement.

10. The authors state the improvement in feed efficiency on day 28-42 by adding lysophospholipids. Still, some researchers would argue that the overall feed efficiency of lysophospholipids is actually 4% higher than control, which may not be economically practical if we take the feed cost per unit of gain into consideration.

11. The title of the manuscript includes health status, but no removal rate, antibiotics-treatment percentage, and mortality was reported in the result.

12. Not sure why 26.68% of crypt cell proliferation in the proximal jejunum of barrows reported in table 4 did not significantly differ from gilts. Previous research has shown that numbers of Ki-67 positive cells positively correlated with villus height and crypt depth in the small intestine of nursery pigs, but it seems like the current study did not observe changes in morphology but observed changes in ki-67 positive cells. An intriguing finding. I also recommended showing representative images of jejunal morphology to replace table 4.

13. Both castrates and male are used in the manuscript. This should be consistent and used the appropriate term. Both female and male is also not appropriate.

13. Ln 300-303: I can't entirely agree with the statement that “pigs display greater ability to adapt minor variations in nutrient content, which in turn might lead to less sensitive response to feed areas additive supplementation.” Many feed additive studies focus on nursery phase, and weaned pigs should easily discern the effectiveness of additives, such as enzymes, flavors, sweeteners, and phytogenic products.

14. Ln 323-325: “The prevalence of diarrhoea in need of treatment from day five to 12 was greater in the LPL group (265 doses) than CON group (200 doses).” This should be reported and analyzed rather than presenting in the discussion section. If the incidence of diarrhea is available, this information should be reported as well.

15. Ln 357-359: I'm afraid I have to disagree that shorter villi is a consequence of the growth check in the first two weeks after weaning. This is because pigs fed LPL were eating well in the first two weeks, and later pigs performed well in both treatments in the last phase, which is the time that intestinal tissue was collected. Therefore, it is hard to state that shorter villi is a consequence of the growth check. The authors also reported the morphology could be a temporary change. This suggests that the intestinal structure should be recovered after a few weeks. So it is harder to correlate the current response is associated with the dietary treatment.

16. Ln 376-386: I strongly disagreed that female had better performance than male is due to “the general earlier physiological maturation as observed in other species.” Indeed, it is well recognized that gilts are associated with lower average daily gain, feed intake with better feed efficiency than barrows, so the actual physiology in barrow allows its faster growth than gilts. The discussion here should be revised or removed.

17. One of the study’s objectives was to analyze the concentrations of IgA in the jejunal mucosa, but the authors did not persuade readers to believe using immunohistochemistry to perform the measurement is the right approach. The authors had difficulty with the quantitative analysis. It is common to use an ELISA kit to determine intestinal mucus secretory IgA. Also, secretory IgA in the ileum is also worth studying since the immunome differences between the ileum and jejunum exist.

Minor comments:

1.  In several places, the authors have used an abbreviation, but it seems they did not define it

2. References are inconsistent. The chapter and pages should be provided if citing from a book

3. The conclusion in the section 5. should be consistent with what have been summarized in the simple summary and the abstract

Author Response

  1. In the introduction, the authors acknowledged that previous results on the application of lysophospholipids are inconsistent due to the dose and source of dietary fat, but they did not design the study to evaluate different inclusion rates of lysophospholipids with multiple lipid sources or at least with animal fats. Instead, they repeated as a previous study to test the supplementation of lysophospholipids at a single dose with soybean oil, which is already known for its great digestibility in young animals.

The perspective of LPL effect on multiple lipid sources have now been removed from the Introduction.

  1. Ln 78-83: the authors provided a couple of studies that evaluated the supplementation of lysophospholipids on growth performance of poultry and swine. They should indicate that the study is based on broiler or swine trials. For example, “There are several examples of improvements in FCR in weaners with dietary addition of LPL [21, 22, 23].” References 21, 22, and 23 are all from poultry, but the authors stated the improvements in FCR in swine. This is considered overgeneralization in scientific writing.

It is now specified that it is in broilers.

  1. Ln 89-93: The significance of this research should be clearly stated. Providing information about tissue morphology and pigs raised under commercial environments is interesting, but how does this information benefit the field or why the research should be conducted in this approach?

Now rephrased to; Thus, the overall objective of this study was to investigate the effect of LPL on growth performance and intestinal morphology of weaners housed under conventional production conditions to evaluate its value as a feed additive in young pigs.

  1. The study used 800 weaned pigs with an initial body weight of 6.96 kg

± 1.58 kg. This creates a coefficient of variation of almost 23%. The high CV suggests no selection on the allotment and potentially with a considerable variation between pens within the treatment that might compromise treatment effects even though the authors balanced the initial body weight with similar average body weight between treatments in the beginning.

We agree with the author, but this illustrates the importance of testing a feed additive under commercial conditions. The CV is not controllable for the pig producer, and thus a feed additive must “perform” regardless of a high or low variation of BW at weaning.

  1. For sampling, 1 pig per pen were randomly selected. As each pen has 25 pigs at the allotment of the current feeding study, the selection of 1 pig without any criteria is hardly a sufficient number to generate meaningful results. Some researchers can argue that your findings are from sampling error.

We agree with the reviewer, however the number pigs used were within the practical limits of this study. Furthermore, it appears that the number of animals used is within the normal range when compared with similar studies. We aimed to select a pig that was closest to the average pen size on a visual basis. 

  1. The experimental diets were 4 phases in table 1, but this differs from what their description in the section 2.2. Also, very little detail is provided in diet composition. While diets are isocaloric with equal SID Lys, the available phosphorus, Ca:P ratio, NDF are missing. Also, no trace minerals content was reported. Please define FU/kg in the table. Analyzed nutrients should also report in the table.

We are not sure which discrepancy between table 4 and section 2.2. Please elaborate, and we will of course correct accordingly. In general, table 4 has now been updated nutrient composition and trace minerals. FU/kg is now defined in footnote 1. In case available phosphorous and Ca:P ration is a prerequisite, please let us know. NDF we do not normally report in Danish pig diet (I think).

  1. For the morphology, the intact villi was measured with the number of villi per slide from 1-17, which is a very small number. Normally, researchers will randomly select 10 fields under 100 × magnification or other magnification to measure villi and crypt. No relevant information was provided in the methods.

This has now been reevaluated among the authors and now rephrased in the manuscript.

  1. What is the reason for not counting the number of Ki-67 positive cells per field and counting the total enterocytes per field to calculate the percentage of cell proliferation?

The chosen method was not sophisticated enough to allow for counting number of Ki-67 positive cells per field, and thus alternatively the percentage of area being constituted by Ki67 positive nuclei was then calculated instead. The method was evaluated by experience researcher in the lab where the method was conducted.

  1. Although the authors stated that supplementation of lysophopholipids can have positive impact on performance by recovering from a poorer starting point, the intake of pigs in lysophopholipid group is actually better than control with a similar growth rate in the first 4 weeks. For me, this suggests that those pigs did not have a hard time post-weaning because pigs were eating well. Hence, it is hard to convince readers of Animal Science field to accept the author’s statement.

Indeed, the ADFI is greater, but the FCR is not, which is why we suggest that the LPL were off to a poor start. In addition, the ADG is lower in the LPL group the first 14 days, which we believe indicate some starting issues, despite a well accommodated weaning transition from a feed intake perspective.

  1. The authors state the improvement in feed efficiency on day 28-42 by adding lysophospholipids. Still, some researchers would argue that the overall feed efficiency of lysophospholipids is actually 4% higher than control, which may not be economically practical if we take the feed cost per unit of gain into consideration.

We completely agree and have stated in the conclusion that “In conclusion, weanling pigs fed LPL supplemented diet did not improve growth performance during the overall experimental period.”, as it is only in the last phase that we observed some minor differences. We did not consider the economic aspect of this study, which is why this was not discussed. 

  1. The title of the manuscript includes health status, but no removal rate, antibiotics-treatment percentage, and mortality was reported in the result.

The word “health” has now been removed from the title.

  1. Not sure why 26.68% of crypt cell proliferation in the proximal jejunum of barrows reported in table 4 did not significantly differ from gilts. Previous research has shown that numbers of Ki-67positive cells positively correlated with villus height and crypt depth in the small intestine of nursery pigs, but it seems like the current study did not observe changes in morphology but observed changes in ki-67 positive cells. An intriguing finding. I also recommended showing presentative images of jejunal morphology to replace table 4.

The 26.68% was a very unfortunate spelling mistake, and the correct number 6.68. We apologize for this sloppy mistake. Images of the jejunal morphology is not available.   

  1. Both castrates and male are used in the manuscript. This should be consistent and used the appropriate term. Both female and male is also not appropriate.

Now only castrates is used.

  1. Ln 300-303: I can't entirely agree with the statement that “pigs display greater ability to adapt minor variations in nutrient content, which in turn might lead to less sensitive response to feed areas additive supplementation.” Many feed additive studies focus on nursery phase, and weaned pigs should easily discern the effectiveness of additives, such as enzymes, flavors, sweeteners, and phytogenic products.

We assume this relates to Ln 284-287 in the current revised manuscript. The phrase was meant as a comparison to broilers, where feed additives often have greater effect than in pigs. Speaking with peers and colleagues there is no apparent explanation for this, but most agree this is the case. There are many non-peer reviewed tests (published in Danish) with a substantial high number of replicates, that show minor if any effect of several feed additives. In fact, only a few feed additives show robust effect of piglet performance.  

  1. Ln 323-325: “The prevalence of diarrhoea in need of treatment from day five to 12 was greater in the LPL group (265 doses) than CON group (200 doses).” This should be reported and analyzed rather than presenting in the discussion section. If the incidence of diarrhea is available, this information should be reported as well.

Only data concerning number of treatments are available, but presented under Results instead. A statistical analysis was not possible.

  1. Ln 357-359: I'm afraid I have to disagree that shorter villi is a consequence of the growth check in the first two weeks after weaning. This is because pigs fed LPL were eating well in the first two weeks, and later pigs performed well in both treatments in the last phase, which is the time that intestinal tissue was collected. Therefore, it is hard to state that shorter villi is a consequence of the growth check. The authors also reported the morphology could be a temporary change. This suggests that the intestinal structure should be recovered after a few weeks. So, it is harder to correlate the current response associated with the dietary treatment.

This has now been rephrased in Ln 335-341.

  1. Ln 376-386: I strongly disagreed that female had better performance than male is due to “the general earlier physiological maturation as observed in other species.” Indeed, it is well recognized that gilts are associated with lower average daily gain, feed intake with better feed efficiency than barrows, so the actual physiology in barrow allows its faster growth than gilts. The discussion here should be revised or removed.

We agree and the sentence has now been removed.

  1. One of the study’s objectives was to analyze the concentrations of IgA in the jejunal mucosa, but the authors did not persuade readers to believe using immunohistochemistry to perform the measurement is the right approach. The authors had difficulty with the quantitative analysis. It is common to use an ELISA kit to determine intestinal mucus secretory IgA. Also, secretory IgA in the ileum is also worth studying since the immune differences between the ileum and jejunum exist.

Everything concerning immunoglobulins has now been removed from the manuscript, as we want to focus only on growth performance and gut morphology.

Minor comments:

  1. In several places, the authors have used an abbreviation, but it seems they did not define it.

Now checked and corrected.

  1. References are inconsistent. The chapter and pages should be provided if citing from a book.

We are not sure exactly which references are in question, but will of course correct then possible.

  1. The conclusion in the section 5. should be consistent with what have been summarized in the simple summary and the abstract.

We believe it is now.